# Self-Assembled Matrine-PROTAC Encapsulating Zinc(II) Phthalocyanine with GSH-Depletion-Enhanced ROS Generation for Cancer Therapy

**DOI:** 10.3390/molecules29081845

**Published:** 2024-04-18

**Authors:** Sitong Lai, Bing Wang, Kunhui Sun, Fan Li, Qian Liu, Xie-An Yu, Lihe Jiang, Lisheng Wang

**Affiliations:** 1School of Medicine, Guangxi University, Nanning 530004, China; laisitong0718@163.com (S.L.); lifanffff@163.com (F.L.); lq622727@126.com (Q.L.); 2NMPA Key Laboratory for Quality Research and Evaluation of Traditional Chinese Medicine, Shenzhen Institute for Drug Control, Shenzhen 518057, China; wangbingszyj@163.com (B.W.); sunkunhuilst@163.com (K.S.); yuxieanalj@126.com (X.-A.Y.); 3School of Basic Medical Sciences, Youjiang Medical University for Nationalities, Baise 533000, China; 4Shenzhen Key Laboratory of Southern Subtropical Plant Diversity, Fairylake Botanical Garden, Shenzhen & Chinese Academy of Science, Shenzhen 518004, China

**Keywords:** matrine, proteolysis-targeting chimeras, nanodrug, reactive oxygen species, glutathione

## Abstract

The integration of a multidimensional treatment dominated by active ingredients of traditional Chinese medicine (TCM), including enhanced chemotherapy and synergistically amplification of oxidative damage, into a nanoplatform would be of great significance for furthering accurate and effective cancer treatment with the active ingredients of TCM. Herein, in this study, we designed and synthesized four matrine-proteolysis-targeting chimeras (PROTACs) (depending on different lengths of the chains named LST-1, LST-2, LST-3, and LST-4) based on PROTAC technology to overcome the limitations of matrine. LST-4, with better anti-tumor activity than matrine, still degrades p-Erk and p-Akt proteins. Moreover, LST-4 NPs formed via LST-4 self-assembly with stronger anti-tumor activity and glutathione (GSH) depletion ability could be enriched in lysosomes through their outstanding enhanced permeability and retention (EPR) effect. Then, we synthesized LST-4@ZnPc NPs with a low-pH-triggered drug release property that could release zinc(II) phthalocyanine (ZnPc) in tumor sites. LST-4@ZnPc NPs combine the application of chemotherapy and phototherapy, including both enhanced chemotherapy from LST-4 NPs and the synergistic amplification of oxidative damage, through increasing the reactive oxygen species (ROS) by photodynamic therapy (PDT), causing an GSH decrease via LST-4 mediation to effectively kill tumor cells. Therefore, multifunctional LST-4@ZnPc NPs are a promising method for killing cancer cells, which also provides a new paradigm for using natural products to kill tumors.

## 1. Introduction

The natural product matrine, a pleiotropic alkaloid isolated from *Radix Sophorae flavescentis* (botanical name *Sophora flavescens*, Chinese name Ku Shen), possesses a wide range of pharmacological properties, including anti-tumor, antiviral, anti-inflammatory, antibacterial, antipyretic, and analgesic properties [1,2,3,4,5,6,7]. Recently, there has been a wealth of research demonstrating that matrine has a cytotoxic effect on cancer cells by inhibiting cancer cell proliferation and metastasis, inducing cell cycle arrest, and accelerating apoptosis [8]. However, despite its multiple pharmacological potentials, the clinical applications of matrine have been restricted due to its large dosage and low pharmacological activity [9,10]. Along the same lines, the short half-life and poor in vivo bioavailability of matrine often limit its actual efficacy [11]. Many researchers have revealed that matrine has very low bioavailability, merely 18.5%, obtained from mice pharmacokinetic studies of matrine [12]. Therefore, given these limitations of matrine itself, there is an urgent need to explore new technical solutions for modifying and optimizing the structure of matrine to acquire more active matrine derivatives.

In recent years, PROTACs have received widespread attention as a promising strategy for anti-cancer research [13]. PROTACs, small molecules comprising ligands for the protein of interest and an E3 ligase-recruiting ligand, induce proteins of interest (POIs) degradation through the ubiquitin–proteasome pathway [14,15]. Compared with traditional occupancy-driven small molecule inhibitors, event-driven protein hydrolysis-targeting chimeras have surprisingly better therapeutic effects for achieving the same therapeutic effect in smaller doses due to their high efficiency and low toxicity [14,16]. Therefore, PROTAC technologies undoubtedly provide inspiration to construct a series of matrine PROTAC units that not only retain the original pharmacological activity of matrine but also significantly improve its therapeutic activity.

The tumor microenvironment (TME), a complex physical and biochemical system including hypoxia, weak acidity, high GSH, and high hydrogen peroxide (H_2_O_2_) content, makes drugs unable to function properly, leading to the survival and progression of cancer cells [17,18,19]. To counter this status, several research groups have enhanced anti-tumor ability by inducing the intracellular GSH depletion of tumor cells [20,21,22,23]. Fortunately, matrine, the major active component of the traditional Chinese medicine *Sophora flavescens*, can reduce the GSH of tumor cells [10]. However, in coping with the complex and ever-changing tumor environment, combination therapy has shown synergistically enhanced anti-cancer effects and lower side effects [24]. Photodynamic therapy (PDT) is a minimally invasive treatment modality that relies on photosensitizers (PSs) to absorb energy at a specific wavelength for producing a large amount of reactive oxygen species (ROS) [25,26,27], further mediating tumor cell necrosis and apoptosis [28]. Ke and co-workers prepared a nanocomplex to enhance chemodynamic therapy by incorporating Fe_3_O_4_ and glucose oxidase into a polyprodrug-based vesicle. Lan and his colleagues also constructed a carrier-free nanodrug made up of gambogic acid, chlorin e6, and folic acid for cooperative cancer treatment. However, the inorganic or metallic properties of nanomaterials like Fe_3_O_4_, as well as the addition of multiple theranostic agents, increase the metabolic load, raising concerns about their potential toxicity to normal tissues [24,29]. Therefore, it is reasonable to deduce that if matrine and photodynamic therapy photosensitizers are integrated into a single nanoparticle, amplifying oxidative stress via a dual modulation of the TME strategy, including increasing ROS levels and reducing GSH levels, which together with chemotherapy to kill cancer cells, this particle would be very significant.

Herein, in our study, we designed and synthesized four matrine derivatives with different linker lengths (LST-1, LST-2, LST-3, LST-4) based on PROTAC technology, detecting that LST-4, with the longest linker, presented the greatest anti-tumor effect. Taking HepG2 cells as an example, LST-4 not only greatly enhances the anti-tumor activity of matrine and possesses extremely low toxicity to normal cells but also preserves the performance of matrine to decrease GSH. In addition, it was unexpectedly discovered that LST-4 could spontaneously assemble into nanostructures (LST-4 NPs) under ultrasound due to the unique amphiphilic structure of hydrophilic matrine linked to lipophilic thalidomide through long polyethylene glycol (PEG) chains, which further promoted the anti-tumor effect and the capacity to reduce GSH. Furthermore, it was discovered through the UV–Vis and fluorescence (FL) spectra that LST-4 NPs could rupture in response to the acidic characteristics of the TME. The cell fluorescence imaging confirmed that LST-4 NPs were indeed taken up by HepG2 cells and highly enriched in lysosomes. Based on the above, LST-4 NPs further successfully loaded poorly soluble ZnPc to obtain LST-4@ZnPc NPs (Figure 1). In the acidic environment of lysosomes, LST-4@ZnPc NPs ruptured to release ZnPc, which increased ROS levels and decreased GSH levels through dual modulation of the TME strategy. Our study, combined with the chemotherapy of LST-4 NPs themselves, goes a step further in improving the cytotoxicity of HepG2 in multiple dimensions. Additionally, the fluorescence properties of LST-4 and ZnPc provide full monitoring of the uptake status of LST-4@ZnPc NPs in cells. Therefore, this study proposes a new functional nanodrug that not only responds to TME for targeting release, but also regulates unfavorable TME through the synergistic effect of the self-integration of oxidative stress, which can be combined with chemotherapy to effectively kill tumor cells.

## 2. Results and Discussion

### 2.1. Synthesis of LST-1, LST-2, LST-3, LST-4

There is extensive evidence in the literature indicating that linkers play a crucial role in the activity of PROTACs [30], specifically that manifesting minor changes in their length and physicochemical properties might lead to drastic changes in the degradation efficiency of target proteins [31]. What was striking, however, was that the minimal change in tyrosine ester-bonded PEG chains not only significantly altered the anti-tumor activity of matrine PROTACs, but also laid the foundation for the preparation of self-assembled PROTAC nanoparticles themselves. In addition, there are reports in the literature indicating that some small-molecule drugs with appropriate water solubility and lipid solubility can self-assemble into nanoparticles to further improve therapeutic effects [32,33]. Therefore, four matrine PROTAC small molecules (LST-1, LST-2, LST-3, and LST-4) were successfully synthesized through a combination of tyrosine esters and PEG chains in this study. The synthesis route was shown in Figure 1, including the structures of various intermediates and target compounds (LST-1, LST-2, LST-3, LST-4), which were characterized by using ^1^H-NMR and ^13^C-NMR; the target compounds were validated via mass (Appendix A).

### 2.2. LST-1, LST-2, LST-3, and LST-4 Inhibited Tumor Cell Proliferation

Given the superiority of PROTAC, the cytotoxicity of matrine PROTAC molecules towards HepG2 cells and L02 cells was measured via the cell counting kit 8 (CCK-8) method to further evaluate the enhanced therapeutic effect of PROTAC technology on matrine. Overall, all four matrine PROTAC molecules showed significant dose-dependent therapeutic effects on HepG2 cells, with the viability of HepG2 cells decreasing as the concentration of the corresponding compound increased (Figure 2a–d). The corresponding IC_50_ values were 162.1 μM, 140.5 μM, 119.6 μM, and 106 μM, respectively, greatly enhancing the anti-tumor activity of matrine (Appendix A). In particular, LST-4, with an IC_50_ value equivalent to 65.59% of LST-1, performed the best among them (Figure 2e). Meanwhile, the four matrine PROTAC molecules exhibited good biosafety in the concentration range of 0–400 μM, with almost no cytotoxicity to L02 cells (Appendix A). Therefore, the enormous potential of the matrine PROTAC molecule in anti-tumor treatment has led to the selection of LST-4 with the best therapeutic effect for subsequent experimental applications to further enhance the anti-tumor efficacy of matrine.

### 2.3. LST-4 Regulated the Levels of p-Akt, p-Erk 

The PI3K/Akt pathway is an intracellular signaling cascade consisting of phosphatidylinositol 3-kinase (PI3K), Akt, and downstream effectors [34,35]. Erk1/2 is a valuable protein consisting of the MAPK/Erk pathway that can drive cell survival, growth, and proliferation [36,37]. Matrine triggers autophagy in glioma cells by blocking the Akt/mTOR signaling pathway and inhibiting the phosphorylation of Akt and mTOR, thereby exerting anti-tumor effects [38]. Additionally, matrine induces autophagy via inhibition of the PI3K/AKT/mTOR pathway and up-regulation of Beclin-1 [39]. Matrine was found to suppress ovarian cancer cell viability, migration and invasion via the p38MAPK-mediated ERK/JNK signaling pathway [40]. Therefore, we speculate that LST-4 (matrine PROTAC) kills tumors by degrading proteins in the p-Erk1/2 and p-Akt signaling pathways. We used protein molecular imprinting experiments to investigate the protein expression of p-Erk1/2 and p-Akt in cells after administration. The results show that LST-4 can downregulate the level of p-Erk1/2 and p-Akt, indicating that LST-4 also exerted an antitumor effect via the p-Erk1/2 and p-Akt signaling pathways (Figure 2f).

### 2.4. Photophysical Properties of LST-4 and Consumption of Intracellular GSH by LST-4

In all of the above experiments, compound LST-4 showed notable effectiveness in killing tumor cells among the four compounds. Hence, the photophysical properties of LST-4, including UV–Vis and fluorescence properties, were verified to further deepen the research on the anti-tumor activity of LST-4. The maximum absorption wavelength of LST-4 was 409 nm (Figure 3a), displaying a new ultraviolet absorption that distinguishes it from matrine. Moreover, LST-4 also exhibited unique fluorescence properties distinct from matrine, with excitation and emission wavelengths of 416 nm and 503 nm, respectively (Figure 3b,d and Appendix A). Both its UV–Vis and FL spectra revealed a good linear relationship between intensity (FL or UV–Vis absorption in MeOH) and their corresponding concentrations (Appendix A), demonstrating good concentration-dependent activity. This also indirectly proved the successful synthesis of matrine PROTAC (LST-4), providing the feasibility for subsequent subcellular localization.

LST-4 is a product of applying PROTAC technology to matrine that inevitably retains the characteristics of matrine, including GSH consumption in tumor cells. Due to this, a detection kit was used to test whether LST-4 could reduce GSH in HepG2 cells. Our results appear to show that LST-4 can deplete GSH in HepG2 cells: the GSH level kept dependently declining with the rising concentration of LST-4 (Figure 3c), indicating that LST-4 also has the ability to consume GSH in tumor cells, laying the groundwork for subsequent oxidative stress strategies targeting tumor cells.

### 2.5. Characterization of LST-4 NPs and LST-4@ZnPc NPs

Self-assembly is a type of nanostructure with significant potential for development and has been widely studied due to its high drug loading, better biocompatibility, and excellent biosafety. Furthermore, numerous studies have shown that assembling monomers into nanoparticles can produce more effective biological activity [41,42]. Here, LST-4 is composed of lipid-soluble thalidomide and water-soluble matrine connected together through a combination of tyrosine esters and PEG chains, allowing LST-4 to meet self-assembly conditions and form nanoparticles without any adjuvants. LST-4 has an amphiphilic structure that can spontaneously self-assemble into nanoparticles (LST-4 NPs) under ultrasound, with the hydrophilic matrine as the outer shell and the lipophilic thalidomide as the inner core. The loaded ZnPc is a poorly soluble compound with low polarity, having a strong interaction with the hydrophobic end of LST-4. Therefore, ZnPc can be encapsulated in the nanocarrier through hydrophobic interaction forces during assembly. Thus, in this study, we constructed an acid-responsive nanoparticle that could consume the GSH of tumor cells, successfully loading the photosensitizer ZnPc to integrate the reduced GSH and increased ROS into one nanoparticle.

The states of ZnPc, LST-4, LST-4 NPs, and LST-4@ZnPc NPs are shown in Figure 4a: they were all uniformly transparent liquids. Among these, ZnPc is an insoluble compound, causing its UV–Vis absorption and fluorescence intensity to gradually decrease with the increase in water ratio in the solvent under the same concentration conditions (Appendix A). The photophysical properties of LST-4 NPs also showed significant changes, including a maximum absorption wavelength of 420 nm and excitation and emission wavelengths of 424 nm and 525 nm (Appendix A), respectively. Additionally, LST-4@ZnPc NPs possess the UV–Vis absorption and fluorescence peaks of LST-4NPs and ZnPc. Both LST-4@ZnPc NPs and LST-4 NPs also exhibited significant redshift in fluorescence and UV–Vis absorption spectra compared to LST-4 monomers (Figure 4b,c), which indicates that LST-4 NPs were successfully synthesized and LST-4@ZnPc NPs loading with ZnPc were also successfully synthesized. On the other hand, the results show that LST-4 NPs were uniformly spherical with an average hydrated particle size of 146 nm and a Zeta potential of −40.82 mV (Figure 4d, Appendix A), indicating the evenly distributed and good stability of the LST-4 NPs. Furthermore, as is evident by TEM and DLS analyses (Figure 4e–g), LST-4@ZnPc NPs displayed spherical shapes with nanosized structures and uniform dispersion, with an average hydration particle size of 169 nm and a Zeta potential of −48.5 mV, directly showing the successful preparation of LST-4 NPs loaded with ZnPc.

The nanosphere solution was measured under different conditions to evaluate the stability of LST-4@ZnPc NPs and LST-4 NPs. LST-4 NPs and LST-4@ZnPc NPs remained in a uniform state in ultrapure water, phosphate buffer solution (PBS), Dulbecco’s Modified Eagle Medium (DMEM), and DMEM containing 50% fetal bovine serum (FBS) for 24 h without precipitation after centrifugation (Appendix A). The LST-4@ZnPc NP solution maintained size stability for 10 days (Appendix A). These results indicate that LST-4@ZnPc NPs have good physiological stability and long-term stability.

In addition, the UV–Vis and fluorescence intensity of LST-4 NPs increased as environmental acidity increased (pH 7.0→3.0) (Figure 4h, i), indicating that LST-4 NPs were prone to deaggregation in an acidic environment. This phenomenon indicates that LST-4 NPs have the characteristic of acid-responsive release in the tumor microenvironment. Due to this, LST-4 NPs can carry ZnPc with a drug loading efficiency of 74.44% (Appendix A). Comparing the cumulative release rate of LST-4@ZnPc NPs in normal dialysate (pH = 7.4) and simulated TME dialysate (pH = 5.0), the cumulative release rate of ZnPc increased from 20.9% to 69.2% (Appendix A), indicating that LST-4 NPs are a carrier with the acid-sensitive property of responding to the acidic environment of the lysosome in tumor cells or the tumor microenvironment, which can target the release of ZnPc inside the tumor.

### 2.6. In Vitro Laser-Triggered Oxidation Regulation by LST-4@ZnPc NPs 

The photodynamic properties of LST-4@ZnPc NPs were evaluated by using 1,3-diphenylisobenzofuran (DPBF) to monitor reactive oxygen species (ROS) generation (Appendix A). ZnPc is a photosensitizer that can generate a large amount of ROS during the irradiation of a 690 nm laser (0.2 W/cm^2^) for 2 min (Figure 5a,b and Appendix A). Correspondingly, LST-4@ZnPc NPs can also produce ROS under the same conditions. Fortunately, when the LST-4@ZnPc NPs were in the acidic environment, the rate of decrease in the DPBF absorbance value at 400–450 nm increased (Figure 5c,d), and the change was more dominant, indicating that LST-4 NPs can indeed respond to the acidic tumor microenvironment to further improve the release of ZnPc for producing more singlet oxygen (^1^O_2_) under 690 nm laser irradiation (Figure 5e,f), which indirectly indicates that the LST-4 NPs successfully loaded ZnPc. During this period, the LST-4 and LST-4 NPs were used as controls under the same conditions.

### 2.7. Cellular Uptake of LST-4 NPs and LST-4@ZnPc NPs

Research has shown that some organic small molecules exhibit spontaneous fluorescence, serving as fluorescent probes for cell imaging [43]. Therefore, the fluorescence inherent in LST-4 NPs can be used to detect uptake of LST-4 NPs and LST-4@ZnPc NPs by cells. Here, the fluorescence emission wavelengths of LST-4NPs and ZnPc are 525 nm and 683 nm with green and purple fluorescence, respectively. LST-4 NPs and LST-4@ZnPc NPs were incubated in HepG2 cells for different times (0, 0.5, 1, 2, and 4 h). As shown in Appendix A, it can be seen that LST-4 NPs showed weak green fluorescence in the cell at 0.5 h, suggesting that LST-4 NPs entered the cells at 0.5 h. The fluorescence intensity gradually increased with incubation time such that obvious green fluorescence appeared in the cell after 4 h, indicating that the LST-4 NPs mostly entered the cells. Similarly, the fluorescence of ZnPc within cells also increased with incubation time (Figure 6), indicating that LST-4@ZnPc NPs could also be uptaken by HepG2, which implies that LST-4 NPs and LST-4@ZnPc NPs have excellent EPR effects in terms of HepG2 cell uptake.

### 2.8. Subcellular Localization of LST-4 NPs and LST-4@ZnPc NPs

To further validate the uptake of LST-4 NPs and LST-4@ZnPc NPs in HepG2 cells, subcellular localizations of LST-4 NPs and LST-4@ZnPc NPs were investigated by inverted fluorescence microscopy. LysoTracker Red, MitoTracker Deep Red, and Hoechst 33342 were used as indicators for lysosomes, mitochondria, and nuclei, represented with red, deep red, and blue, respectively, to avoid interference from fluorescent channels. As shown in Appendix A, the green fluorescence of LST-4 NPs overlapped with the red fluorescence of lysosomes, producing yellow-green fluorescence. Likewise, it was noteworthy that the green fluorescence of LST-4 NPs almost overlapped with the purple fluorescence from ZnPc to appear as dark green fluorescence spots (Figure 7). It then overlapped with the red fluorescence of lysosomes to produce the red-orange fluorescence spots, indicating that ZnPc was successfully loaded into the LST-4 NPs, accumulating effectively in the cancer cells and targeting lysosomal compartments.

### 2.9. Intracellular ROS Generation via LST-4@ZnPc NPs and Intracellular GSH Depletion by LST-4 NPs

To more clearly explore whether the generation of ROS comes from LST-4@ZnPc NPs, here, cellrox^TM^ orange was used to further measure intracellular ROS production from the PDT of NPs. ZnPc is a photosensitizer with excellent performance, able to generate a large amount of ROS under 690 nm laser irradiating. Nevertheless, the solubility of ZnPc is extremely low, especially since ZnPc will precipitate when it comes into contact with aqueous solutions, resulting in significant difficulties in direct application. Therefore, LST-4 NPs were used as carriers to load ZnPc in this study with the expectation of responding to the acidic environment of tumors to release drugs. As shown in the previous results, the obvious orange fluorescence of cellrox^TM^ orange in cells co-incubated with LST-4@ZnPc NPs was observed only under irradiation with 690 nm (Figure 8a). On the contrary, all other groups did not exhibit fluorescence, especially the control group treated with vitamin C, which did not produce the corresponding fluorescence of reactive oxygen species, proving the authenticity of LST-4@ZnPc NPs producing a large amount of reactive oxygen species under illumination. The above well demonstrates that LST-4 NPs successfully load ZnPc and produce reactive oxygen species under 690 nm laser irradiating. Taken together, these results further indicate that LST-4@ZnPc NPs successfully combine oxidative stress strategies with chemotherapy to kill tumor cells.

Fortunately, LST-4 itself has the function of depleting GSH in tumor cells. It seemed that the co-incubation of LST-4 NPs with HepG2 cells further reduced GSH in tumor cells compared to the same concentration of LST-4 (Figure 8b). Therefore, when LST-4@ZnPc NPs produce reactive oxygen species under light, they can still effectively reduce excess GSH in tumor cells, leading to a lower antioxidant capacity of the tumor cells, which further amplifies oxidative stress to kill the tumor cells.

### 2.10. In Vitro Inhibition of HepG 2 Cells by LST-4 NPs and LST-4@ZnPc NPs

There are a substantial number of reports in the literature suggesting that the self-assembly of small molecule monomers into nanostructures is capable of further enhancing anti-tumor activity [44,45]. Therefore, the anti-tumor activity of LST-4 NPs was further detected using the CCK-8 assay kit. The results revealed that the cell viability of HepG2 decreased with increasing concentration of LST-4 NPs, exhibiting a significant dose-dependent therapeutic effect with an IC_50_ of 47.32 μg/mL (Figure 8c and Appendix A), comparable to the 42.19% of LST-4. These results indicate that self-assembly can actually further improve biological activity in killing tumor cells.

Combined with the results above, it has been proven that LST-4@ZnPc NPs did indeed integrate functions including elevated ROS from PDT and decreased GSH into a single nanoparticle, leading to amplified oxidative stress combined with chemotherapy to more effectively kill tumor cells. Due to the low solubility of ZnPc, it is difficult for ZnPc to enter cells, resulting in the extremely low cytotoxicity of ZnPc to HepG2 cells under light irradiation (Appendix A). However, the drug loading ability of LST-4 NPs enhanced the bioavailability of ZnPc. These results show that the IC_50_ of LST-4@ZnPc NPs in the dark is 43.65 ug/mL, which is almost the same as that of LST-4 NPs. Nevertheless, the IC_50_ value of LST-4@ZnPc NPs declined to 23.45 ug/mL after 5 min of irradiation with a 690 nm laser (0.2W/cm^2^) (Figure 8d,e, Appendix A), indicating LST-4@ZnPc NPs not only possess the chemotherapy effect of LST-4 NPs, but also can deplete GSH with light-mediated controlled release of ROS from ZnPc for achieving synergistic therapeutic effects. The above results show that the successful preparation of LST-4@ZnPc NPs and the idea of their combined therapy was directly demonstrated through cytotoxicity studies in vitro.

## 3. Materials and Methods

### 3.1. Materials and Agents

All chemical materials were reagent-grade and the solvents were analytical-grade, able to be used without further purification except for special requirements for drying and degassing. RPMI1640 media (Gibco, Grand Island, NY, USA), fetal bovine serum (FBS, Gibco, Grand Island, NY, USA), phosphate-buffered saline (PBS, Gibco, Grand Island, NY, USA) and 0.25% Trypsin (Gibco, Grand Island, NY, USA) medium were used for cell culture. The Cell Counting Kit-8 (CCK-8) was obtained from Shanghai Biyuntian Biotechnology Co., Ltd, Shanghai, China. CellROX^TM^ Orange, LysoTracker Red, MitoTracker Green and Hoechst 33342 were purchased from Invitrogen (ThermoFisher Scientific, Waltham, MA, USA). Unless otherwise stated, all materials were acquired from the commercial supplier Aladdin Biochemical Technology Co., Ltd. (Shanghai, China), Anhui Zesheng Technology Co., Ltd. (Anhui, China) and Bide Pharmatech Ltd. (Shanghai, China). Ultrapure water was prepared using a Millipore Simplicity System (Millipore, Bedford, MA, USA). 

### 3.2. Apparatus

The ^1^H-NMR spectra and ^13^C-NMR spectra were recorded with a Bruker Ascend AVANCE III 600 MHz spectrometer (Billerica, MA, USA) or AVANCE III 500 MHz. Mass spectra (MS) were performed using a SCIEX X500R QTOF mass spectrometer (AB.Sciex, Framinghan, MA, USA). The morphology of the nanoparticles was examined with a JEOL JEM 2100Plus Transmission Electron Microscope (JEOL, Tokyo, Japan). A dynamic light scattering apparatus (ZS90, Malvern Instruments, Ltd., Worcestershire, UK) was utilized to determine the size distribution and zeta potential. Cell FL images were captured with a fluorescent microscope (Axio Observer 7, Carl Zeiss, Oberkochen, Germany). 

### 3.3. Synthesis

The following reaction steps mainly refer to previous work and were partially modified [46,47,48]. A mixture of 2-(2,6-dioxopiperidin-3-yl)-4-fluoroisoindoline-1,3-dione (0.83 g, 3 mmol, 1 eq), *N*,*N*-diisopropylethylamine (0.66 g, 5.1 mmol, 1.7 eq), and N_3_-PEGn-CH_2_CH_2_NH_2_ (0.79 g, 3 mmol, 1 eq) in *N*,*N*-dimethylformamide (8 mL) was stirred at 90 °C for 4 h. Then, the reaction system was stirred with ice water (8 mL) and the mixture was extracted with the appropriate amount of ethyl acetate. The ethyl acetate layer was combined and dried with anhydrous Na_2_SO_4_. Next, the solvent was evaporated using a rotary evaporator to obtain the crude product. The crude product was purified by silica gel column chromatography using petroleum ether and ethyl acetate (6:1, *v*/*v*) as the eluent and dried under vacuum to obtain the desired compound, which was product compound 5, including **5a** (0.41 g, 35.3%), **5b** (0.56 g, 41.4%), **5c** (0.60 g, 41.9%) and **5d** (0.66g, 42.6%). The structures and synthesis methods of compounds **5a**, **5b**, **5c**, and **5d** have been reported in multiple articles [46,47].

**5a**: ^1^H-NMR (500 MHz, Chloroform-d) δ 8.31 (s, 1H, OCNHCO), 7.49 (dd, *J* = 8.5, 7.1 Hz, 1H, Ph-H), 7.10 (d, *J* = 7.1 Hz, 1H, Ph-H), 6.94 (d, *J* = 8.5 Hz, 1H, Ph-H), 4.92 (dd, *J* = 12.3, 5.3 Hz, 1H, OCCHN), 3.69 (dt, *J* = 23.7, 5.3 Hz, 4H, CH_2_OCH_2_), 3.50 (t, *J* = 5.4 Hz, 2H, NCH_2_), 3.40 (t, *J* = 5.0 Hz, 2H, CH_2_N_3_), 2.90–2.68, 2.16–2.06 (m, m; 4H, OCCH_2_CH_2_C).

**5b**: ^1^H-NMR (500 MHz, Chloroform-d) δ 8.52 (s, 1H, OCNHCO), 7.47–7.45 (m, 1H, Ph-H), 7.08 (d, *J* = 7.1 Hz, 1H, Ph-H), 6.91 (d, *J* = 8.5 Hz, 1H, Ph-H), 6.48 (t, *J* = 5.7 Hz, 1H, PhNH), 4.95–4.84 (m, 1H, OCCHN), 3.74–3.64 (m, 8H, CH_2_OCH_2_CH_2_OCH_2_), 3.46 (q, *J* = 5.5 Hz, 2H, NCH_2_), 3.36 (t, *J* = 5.1 Hz, 2H, CH_2_N_3_), 2.88–2.67, 2.09 (m, ddd, *J* = 12.2, 5.8, 3.1 Hz; 4H, OCCH_2_CH_2_C).

**5c**: ^1^H-NMR (500 MHz, Chloroform-d) δ 8.03 (s, 1H, OCNHCO), 7.49 (q, *J* = 6.6, 5.5 Hz, 1H, Ph-H), 7.11 (d, *J* = 7.1 Hz, 1H, Ph-H), 6.93 (d, *J* = 8.5 Hz, 1H, Ph-H), 6.49 (s, 1H, Ph-NH), 4.91 (dd, *J* = 12.1, 5.3 Hz, 1H, OCCHN), 3.74–3.63 (m, 12H, CH_2_OCH_2_CH_2_OCH_2_CH_2_OCH_2_), 3.53–3.32 (m, 4H, NCH_2_, CH_2_N_3_), 2.88–2.69, 2.17–2.07 (m, m; 4H, OCCH_2_CH_2_C).

**5d**: ^1^H-NMR (500 MHz, Chloroform-d) δ 8.58 (s, 1H, OCNHCO), 7.47 (dd, *J* = 8.6, 7.1 Hz, 1H, Ph-H), 7.07 (d, *J* = 7.1 Hz, 1H, Ph-H), 6.91 (d, *J* = 8.5 Hz, 1H, Ph-H), 6.47 (t, *J* = 5.7 Hz, 1H, Ph-NH), 4.90 (dd, *J* = 12.2, 5.4 Hz, 1H, OCCHN), 3.71–3.64 (m, 16H, CH_2_OCH_2_CH_2_OCH_2_CH_2_OCH_2_CH_2_OCH_2_), 3.48–3.35 (m, 4H, NCH_2_, CH_2_N_3_), 2.87–2.67, 2.09 (m, ddd, *J* = 9.6, 5.0, 2.3 Hz; 4H, OCCH_2_CH_2_C).

Aniline (2.09 g, 8.05 mmol, 2 eq) and tetrahydrofuran (THF) (10 mL, 0.4 M) were added into a 25 mL three-neck round-bottom flask. We cooled the resulting solution to −40 °C. Then, we added n-BuLi (7.1 mL, 1.13 M solution in hexane, 8.05 mmol, 2 eq). The reaction mixture was stirred at −40 °C for 1 h. Then, 1.0 g (4.03 mmol) of matrine in THF (2 mL) was added to the flask, subsequently raised to 0 °C and stirred for 1 h. Next, ZnCl_2_ (14.00 mL of a 0.86 M solution in THF, 12.08 mmol, 3.00 eq) was added to the system, followed by stirring the reaction mixture for 0.5h. A pre-prepared solution of [Pd (allyl) Cl]_2_ (74 mg, 0.20 mmol, 5%) and allyl acetate (522 μL, 4.83 mmol, 1.2 eq) in THF (2.00 mL) was added. After stirring for 0.5 h, we slowly raised the temperature of the system to 60 °C and reacted it for 10 h. After the reaction is completed, we quenched the mixture with methanol under ice bath conditions, concentrating the liquid under reduced pressure to obtain the crude product. The crude product was purified by silica gel column chromatography using dichloromethane (CH_2_Cl_2_) and ethyl acetate (1:1, *v*/*v*) as the eluent, and dried under vacuum to obtain the desired compound **6** (0.74 g, 74.5%). ^1^H-NMR (600 MHz, Chloroform-d) δ 6.46–6.42 (m, 1H, OCC = CH), 5.87 (dd, *J* = 9.8, 2.0 Hz, 1H, OCCH = C), 4.11 (dd, *J* = 13.0, 4.7 Hz, 1H, NCHC(C)_2_), 3.96 (ddd, *J* = 10.7, 9.2, 6.7 Hz, 1H, NCH(C)_2_), 3.14 (t, *J* = 12.8 Hz, 1H, NCHC(C)_2_), 2.83–2.75 (m, 2H, NCH_2_CCC), 2.58 (dddd, *J* = 18.1, 6.7, 4.9, 1.6 Hz, 1H, NCH(C)_2_), 2.21–2.07 (m, 2H, NCH_2_CCC), 1.92 (ddd, *J* = 14.1, 11.2, 2.9 Hz, 2H, C = CCH_2_), 1.80–1.41 (m, 10H, N(CCH_2_CH_2_CH)_2_C); ^13^C-NMR (126 MHz, Chloroform-d) δ 165.71, 137.49, 124.55, 63.46, 57.26, 51.43, 41.97, 41.49, 34.59, 27.73, 27.35, 26.57, 21.05, 20.71.

Compound **6** (0.31 g, 1.25 mmol, 1 eq) and glycine (0.19 g, 2.5 mmol, 2 eq) were added to the water (2 mL). The reaction mixture was stirred at 80 °C for 12 h. After the completion of the reaction, the solvents were removed by evaporation under reduced pressure. Then, the pure compounds were obtained through recrystallization from an ethanol–water system, which was the product compound **7** (0.28 g, 69.8%). ^1^H-NMR (600 MHz, deuterium oxide) δ 4.38 (dd, *J* = 14.0, 4.6 Hz, 1H, NCHC(C)_2_), 3.81 (ddd, *J* = 12.4, 7.3, 5.8 Hz, 1H, NCH(C)_2_), 3.54 (s, 3H, NHCH_2_C = O), 3.31 (d, *J* = 6.2 Hz, 2H, OCCH_2_C), 3.29–3.28 (m, 1H, (C)_2_CHN), 2.96–2.68 (m, 4H, N(CH_2_CCC)_2_C), 2.45 (dd, *J* = 17.8, 5.5 Hz, 1H, NCHC(C)_2_), 2.23–2.18 (m, 1H, NCH(C(C)_2_)_2_), 2.07–1.96 (m, 4H, NCCH_2_CC, CH(C)_2_CCH(C)_2_), 1.84–1.65 (m, 8H, N(CCH_2_CH_2_C)_2_C); ^13^C-NMR (126 MHz, deuterium oxide) δ 172.40, 169.92, 63.27, 55.70, 49.98, 48.95, 47.65, 41.40, 41.01, 40.14, 35.92, 33.68, 28.45, 25.10, 23.77, 18.98, 18.55.

Boc-L-tyrosine methyl ester (5.90 g, 20 mmol, 1 eq) and potassium carbonate (5.53 g, 40 mmol, 2 eq) were added to the solution of propargyl bromide (4.76 g, 40 mmol, 2 eq) in anhydrous *N*,*N*-dimethylformamide (DMF) (15 mL). The reaction mixture was stirred at room temperature for 6 h. Then, the reaction system was stirred with ice water (200 mL) and the mixture was extracted with ethyl acetate (200 mL). The ethyl acetate layers were combined and dried with anhydrous Na_2_SO_4_. Afterwards, the solvent was evaporated using a rotary evaporator to obtain the crude product. The crude product was purified by silica gel column chromatography using petroleum ether and ethyl acetate (5:1, *v*/*v*) as the eluent and was dried under vacuum to obtain the desired compound as a solid in a 97.9% yield (6.53 g), which was the product compound **8**. ^1^H-NMR (600 MHz, Chloroform-d) δ 7.04 (d, *J* = 8.3 Hz, 2H,H-Ph-H), 6.91–6.87 (m, 2H, H-Ph-H), 4.98 (d, *J* = 8.2 Hz, 1H, OCCHN), 4.65 (d, *J* = 2.4 Hz, 2H, OCH_2_C≡C), 4.57–4.48 (m, 1H, NH), 3.70 (s, 3H, OCH_3_), 3.05, 2.99 (dd, *J* = 14.0, 5.8 Hz; dd, *J* = 14.0, 6.2 Hz; 2H, PhCH_2_C), 2.51 (t, *J* = 2.4 Hz, 1H, HC≡C), 1.41 (s, 9H, Boc); ^13^C-NMR (151 MHz, Chloroform-d) δ 172.40, 156.66, 155.10, 130.33, 129.00, 114.97, 79.91, 78.58, 75.54, 55.81, 54.50, 52.21, 37.48, 28.31. 

Compound **8** was placed in a solution of ethyl acetate and hydrochloride (1:1, *v*/*v*) and stirred overnight at room temperature to remove Boc groups. Thin-layer chromatography was used to detect the reaction results during the reaction procedure. After the reaction was completed, the solvent of the reaction mixture was removed to obtain compound **9**, which was a white solid and directly used for the next step of the reaction.

A mixture of compound **7** (0.32 g, 1 mmol, 1 eq) and compound **9** (0.23 g, 1 mmol, 1 eq) in anhydrous dichloromethane (50 mL) was stirred at 0 °C in an ice bath. After the reactant was completely dissolved, 1-ethyl-3-(3-dimethylaminopropyl) carbodiimide hydrochloride (EDCI) (0.23 g, 1.2 mmol, 1.2 eq), 1-hydroxybenzotriazole (HOBT) (0.16 g, 1.2 mmol, 1.2 eq), and *N*,*N*-diisopropylethylamine (DIPEA) (0.16 g, 1.2 mmol, 1 eq) were sequentially added to the reaction mixture, which was subsequently was stirred for 24 h at room temperature. After the reaction was completed, the reaction system was stirred with 50 mL of saturated aqueous NaCl and the mixture was extracted with dichloromethane. The dichloromethane layers were combined and dried with anhydrous Na_2_SO_4_. Afterwards, the solvent was evaporated using a rotary evaporator to obtain the crude product. The crude product was purified by silica gel column chromatography using methanol and dichloromethane (1:20, *v*/*v*) as the eluent and dried under vacuum to obtain the desired compound as a yellow solid in a 48% yield (0.258 g), which was the product compound **10**. ^1^H-NMR (600 MHz, Chloroform-d) δ 7.50 (dd, *J* = 21.8, 8.0 Hz, 1H, OCNHC), 7.04 (dd, *J* = 8.7, 2.4 Hz, 2H, H-Ph-H), 6.89 (dd, *J* = 8.7, 6.9 Hz, 2H, H-Ph-H), 4.78 (dddd, *J* = 18.9, 7.7, 6.7, 5.7 Hz, 1H, OCCHN), 4.66 (dd, *J* = 5.1, 2.4 Hz, 2H, OCH_2_C≡C), 4.32 (dd, *J* = 12.7, 4.3 Hz, 1H, NCHC(C)_2_), 3.71 (d, *J* = 5.5 Hz, 3H, OCH_3_), 3.34–3.21 (m, 2H, OCCH_2_N), 3.12–3.01 (m, 3H, Ph-CH_2_C, C≡CH), 2.85–2.75 (m, 3H, OCCH_2_C, NCH(C)_2_), 2.54, 2.46 (dt, *J* = 4.9, 2.4 Hz; ddd, *J* = 17.0, 4.8, 1.2 Hz; 2H, NCH_2_CC), 2.23–2.17 (m, 1H, NCHC(C)_2_), 2.12 (s, 1H, CNHCO), 1.96 (qd, *J* = 13.8, 12.8, 8.8 Hz, 3H, NCH_2_CC, NCH(C)_2_), 1.82–1.78, 1.74–1.63, 1.54–1.36 (m, m, m; 13H, NCH(C(C)_2_)_2_, NCCH_2_CC, CH(C)_2_CCH(C)_2_, N(CCH_2_CH_2_C)_2_C); ^13^C NMR (151 MHz, Chloroform-d) δ 171.94, 171.42, 171.19, 167.55, 166.92, 156.72, 130.17, 128.82, 115.03, 78.56, 75.65, 63.83, 57.19, 55.82, 52.73, 52.32, 51.58, 50.88, 50.26, 49.68, 48.89, 42.33, 41.81, 38.85, 36.97, 35.56, 30.60, 27.67, 21.12, 20.64.

A mixture of compound **5a** (0.26 g, 0.5 mmol, 1 eq) and compound **10** (0.27 g, 0.5 mmol, 1 eq) in tetrahydrofuran (2 mL) was stirred at room temperature. Then, a brown aqueous solution (0.5 mL) containing copper sulfate (0.15 g, 0.5 mmol, 1 eq) and sodium ascorbate (0.20 g, 1 mmol, 2 eq) was added dropwise to the reaction solution, which was subsequently stirred for 10 h at room temperature. After the reaction was completed, the solvent was removed in a rotary evaporator to obtain the crude product. The crude product was purified by silica gel column chromatography using MeOH and CH_2_Cl_2_ (1:15, *v*/*v*) as the eluent and was dried under vacuum to obtain the desired compound as a yellow solid in a 23.8% yield (0.11 g), which was the product matrine derivative **LST-1**. ^1^H-NMR (600 MHz, DMSO-d6) δ 11.09 (s, 1H, OCNHCO), 8.14 (d, *J* = 1.6 Hz, 1H, Ph-NH), 7.56 (dd, *J* = 8.5, 7.1 Hz, 1H, Triazole-H), 7.11 (dd, *J* = 8.4, 4.0 Hz, 3H, Ph-H), 7.03 (d, *J* = 7.0 Hz, 1H, Ph-H), 6.91 (dd, *J* = 8.8, 2.7 Hz, 2H, Ph-H), 6.59 (t, *J* = 5.9 Hz, 1H, Ph-H), 5.75 (s, 1H, OCNHC), 5.06–5.00 (m, 3H, Triazole-CH_2_-O, NCH(CO)C), 4.57–4.46 (m, 3H, Ph-CCH(CO)N, NCOCH_2_N), 4.15–4.07 (dd, 1H, NCHC(C)_2_), 3.85 (t, *J* = 5.2 Hz, 2H, Triazole-CH_2_), 3.65–3.41 (m, 9H, Ph-NCH_2_CH_2_OCH_2_, OCH_3_), 3.10 (d, *J* = 25.3 Hz, 2H, Ph-CH_2_C), 3.00–2.84 (m, 4H, OCCNHCH(C)CCH(C)N(C)CH), 2.71 (d, *J* = 31.2 Hz, 2H, OCCH_2_C), 2.59–2.51 (m, 2H,OCCCH_2_), 2.34 (dd, *J* = 35.3, 17.5 Hz, 1H, NCH(C)_2_), 2.06–1.98 (m, 2H, OCCH_2_C), 1.86–1.74 (m, 4H, N(CH_2_)_2_), 1.62–1.48, 1.34–1.22 (m,m; 12H, N(CCH_2_CH_2_CH)_2_, CCH_2_C(C)N); ^13^C-NMR (151 MHz, DMSO-d6) δ 173.25, 172.37, 172.29, 170.52, 169.40, 167.75, 157.36, 146.85, 143.10, 136.69, 132.55, 130.58, 130.56, 129.56, 125.31, 117.93, 114.89, 114.87, 111.18, 109.71, 69.37, 69.13, 61.47, 61.44, 57.13, 57.01, 56.22, 55.38, 53.66, 52.38, 51.36, 49.85, 49.03, 42.03, 38.70, 37.86, 36.36, 35.54, 34.38, 31.44, 29.49, 22.70,22.61, 21.20, 20.64. HRMS (ESI): *m/z* calcd. for C_47_H_59_N_10_O_10_ [M+H]^+^ 923.4416, found 923.4418.

Using the same procedure as above, the products obtained by reacting 5b, 5c, and 5d with compound 10 were purified by silica gel column chromatography to obtain **LST-2** (0.18 g, 37.2%), **LST-3** (0.15 g, 29.7%), and **LST-4** (0.22g, 41.7%), respectively.

**LST-2**: ^1^H-NMR (600 MHz, DMSO-d6) δ 11.09 (s, 1H, OCNHCO), 8.15 (d, *J* = 3.2 Hz, 1H, Ph-NH), 7.56 (dd, *J* = 8.6, 7.1 Hz, 1H,Triazole-H), 7.13–7.08 (m, 3H, Ph-H), 7.03 (d, *J* = 7.1 Hz, 1H, Ph-H), 6.93–6.88 (m, 2H, Ph-H), 6.58 (t, *J* = 5.8 Hz, 1H, Ph-H), 5.75 (s, 1H, OCNHC), 5.08–5.02 (m, 3H, Triazole-CH_2_-O, NCH(CO)C), 4.54–4.46 (m, 3H, Ph-CCH(CO)N, NCOCH_2_N), 4.15–4.07 (dd, 1H, NCHC(C)_2_), 3.82 (t, *J* = 5.2 Hz, 2H,Triazole-CH_2_), 3.63–3.40 (m, 13H, Ph-NCH_2_CH_2_OCH_2_CH_2_OCH_2_, OCH_3_), 3.15–3.03 (m, 2H, Ph-CH_2_C), 3.00–2.85 (m, 4H, OCCNHCH(C)CCH(C)N(C)CH), 2.74–2.65 (m, 2H, OCCH_2_C), 2.61–2.50 (m, 2H, OCCCH_2_), 2.39–2.27 (m, 1H, NCH(C)_2_), 2.07–1.97 (m, 2H, OCCH_2_C), 1.84,1.79–1.72 (t, *J* = 10.5 Hz; m; 4H, N(CH_2_)_2_), 1.63–1.48, 1.35–1.22 (m; m; 12H, N(CCH_2_CH_2_CH)_2_, CCH_2_C(C)N); ^13^C NMR (151 MHz, DMSO-d6) δ 173.26, 172.39, 172.37, 170.54, 169.44, 167.75, 157.35, 146.85, 143.05, 136.69, 132.54, 130.57, 130.54, 129.57, 125.24, 117.87, 114.88, 114.86, 111.17, 109.72, 70.06, 70.05, 69.29, 69.21, 63.71, 63.46, 61.50, 57.13, 57.01, 55.38, 53.65, 53.57, 52.37, 49.91, 49.03, 48.16, 42.14, 38.68, 36.36, 36.24, 35.55, 31.45, 30.22, 27.92, 26.57, 22.61, 21.19, 20.67. HRMS (ESI): *m/z* calcd. for C_49_H_63_N_10_O_11_ [M+H]^+^ 967.4678, found 967.4668.

**LST-3**: ^1^H-NMR (600 MHz, DMSO-d6) δ 11.09 (s, 1H, OCNHCO), 8.16 (s, 1H, Ph-NH), 7.57 (dd, *J* = 8.6, 7.0 Hz, 1H,Triazole-H), 7.14–7.08 (m, 3H, Ph-H), 7.03 (d, *J* = 7.0 Hz, 1H, Ph-H), 6.95–6.90 (m, 2H, Ph-H), 6.58 (t, *J* = 5.8 Hz, 1H, Ph-H), 5.75 (s, 1H, OCNHC), 5.05 (d, *J* = 13.0 Hz, 3H, Triazole-CH_2_-O, NCH(CO)C), 4.50 (dt, *J* = 9.8, 5.3 Hz, 3H, Ph-CCH(CO)N, NCOCH_2_N), 4.14–4.04 (dd, 1H, NCHC(C)_2_), 3.80 (t, *J* = 5.2 Hz, 2H,Triazole-CH_2_), 3.68–3.40 (m, 17H, Ph-NCH_2_CH_2_OCH_2_CH_2_OCH_2_CH_2_OCH_2_, OCH_3_), 3.16–3.02 (m, 2H, Ph-CH_2_C), 3.02–2.83 (m, 4H, OCCNHCH(C)CCH(C)N(C)CH), 2.70 (dd, *J* = 30.3, 10.6 Hz, 2H, OCCH_2_C), 2.63–2.51 (m, 2H, OCCCH_2_), 2.31 (dd, *J* = 16.7, 4.4 Hz, 1H, NCH(C)_2_), 2.07–1.98 (m, 2H, OCCH_2_C), 1.84,1.79–1.71 (dd, *J* = 16.9, 9.0 Hz; m; 4H, N(CH_2_)_2_), 1.54, 1.34–1.23 (dd, *J* = 25.7, 10.7 Hz; m; 12H, N(CCH_2_CH_2_CH)_2_, CCH_2_C(C)N); ^13^C NMR (151 MHz, DMSO-d6) δ 173.28, 172.39, 172.37, 170.54, 169.41, 167.76, 157.36, 146.86, 143.04, 136.69, 132.55, 130.58, 130.56, 129.60, 125.26, 117.90, 114.89, 114.87, 111.15, 109.70, 70.32, 70.15, 70.01, 69.32, 69.13, 63.72, 61.52, 61.49, 57.12, 57.02, 55.38, 53.65, 52.37, 50.07, 49.87, 49.02, 48.16, 42.15, 38.70, 38.10, 36.36, 31.45, 30.25, 27.93, 26.58, 22.61, 22.56, 21.11, 20.68. HRMS (ESI): *m/z* calcd. for C_51_H_67_N_10_O_12_ [M+H]^+^ 1011.4940, found 1011.4938. 

**LST-4**: ^1^H-NMR (600 MHz, DMSO-d6) δ 11.09 (s, 1H, OCNHCO), 8.16 (d, *J* = 3.1 Hz, 1H, Ph-NH), 7.57 (dd, *J* = 8.5, 7.1 Hz, 1H,Triazole-H), 7.15–7.09 (m, 3H, Ph-H), 7.04 (d, *J* = 7.0 Hz, 1H, Ph-H), 6.93 (dd, *J* = 8.8, 2.7 Hz, 2H, Ph-H), 6.59 (t, *J* = 5.9 Hz, 1H, Ph-H), 5.75 (s, 1H, OCNHC), 5.09–5.02 (m, 3H, Triazole-CH_2_-O, NCH(CO)C), 4.55–4.47 (m, 3H, Ph-CCH(CO)N, NCOCH_2_N), 4.12 (dd, *J* = 9.4, 5.7 Hz, 1HNCHC(C)_2_), 3.80 (t, *J* = 5.2 Hz, 2H,Triazole-CH_2_), 3.64–3.42 (m, 21H, Ph-NCH_2_CH_2_OCH_2_CH_2_OCH_2_CH_2_OCH_2_CH_2_OCH_2_, OCH_3_), 3.17–3.05 (m, 2H, Ph-CH_2_C), 3.01–2.85 (m, 4H, OCCNHCH(C)CCH(C)N(C)CH), 2.69 (d, 2H, OCCH_2_C), 2.63–2.52 (m, 2H, OCCCH_2_), 2.40–2.28 (m, 1H, NCH(C)_2_), 2.09–1.98 (m, 2H, OCCH_2_C), 1.89, 1.80–1.73 (dd, *J* = 16.2, 11.8 Hz; m; 4H, N(CH_2_)_2_), 1.55, 1.43–1.22 (q, *J* = 12.8 Hz; m; 12H, N(CCH_2_CH_2_CH)_2_, CCH_2_C(C)N); ^13^C NMR (151 MHz, DMSO-d6) δ 173.27,172.36, 171.81, 170.53, 169.41, 167.76, 157.37, 146.87, 143.04, 136.69, 132.56, 130.58, 130.56, 129.58, 125.28, 117.91, 114.90, 114.88, 111.15, 109.71, 70.28, 70.24, 70.22, 70.07, 69.99, 69.34, 69.14, 63.69, 63.43, 61.51, 61.49, 57.05, 56.93, 55.38, 53.68, 53.62, 52.37, 50.18, 49.87, 49.03, 48.18, 42.16, 38.52, 36.35, 36.24, 35.45, 31.45, 30.10, 27.81, 22.61, 21.04, 20.57. HRMS (ESI): *m/z* calcd. for C_53_H_71_N_10_O_13_ [M+H]^+^ 1055.5202, found 1055.5202.

### 3.4. UV–Vis Absorption Spectra of LST-4 NPs

Six test concentrations (100 μg/mL, 80 μg/mL, 60 μg/mL, 40 μg/mL, 20 μg/mL, and 10 μg/mL) were prepared using the stepwise dilution method with MeOH. The UV–Vis absorption spectra of LST-4 in corresponding solvents at room temperature were measured with a UV-2450 spectrophotometer (Shimadzu, Kyoto, Japan) using the corresponding solvents as blank references.

### 3.5. FL Spectra of LST-4 NPs

The test solutions of compound LST-4 at different concentrations (2.5 μg/mL, 2.0 μg/mL, 1.5 μg/mL, 1.0 μg/mL, 0.5 μg/mL, and 0.25 μg/mL) were prepared with MeOH using the stepwise dilution method. The FL emission spectra of LST-4 at room temperature were collected at excitation wavelengths of 416 nm by using a Shimadzu RF 5301PC spectrofluorometer (Shimadzu, Kyoto, Japan) with the slit settings as Ex = 5.0 and Em = 5.0; the emission wavelength was 503 nm. 

### 3.6. Preparation of LST-4 NPs

In a typical procedure [49], the LST-4 was dissolved in MeOH at a concentration of 2 mg/mL to form the mother liquor. Then, the mother liquor (500 μL) was slowly injected into 5 mL of ultrapure water over the last half hour, and ultrasound continued for an additional 2 h at room temperature. During the ultrasonic process, MeOH was removed by N_2_ bubbling above the water surface at a constant rate. After completion, the supernatant was taken for further experiments after centrifugation at 3000 rpm for 10 min. The LST-4 NPs were characterized using DLS and TEM.

### 3.7. Preparation of LST-4@ZnPc NPs

The LST-4 NPs were prepared as templates for synthesizing the LST-4@ZnPc NPs. In this synthetic scheme, LST-4@ZnPc NPs were synthesized using the same procedure described above for the synthesis of LST-4 NPs, except for the addition of the appropriate volume of ZnPc stock solution (0.0875 mg/mL) in THF. Afterward, the supernatant was taken for further experiments after centrifugation at 3000 rpm for 10 min.

### 3.8. Characterization

The LST-4 NPs and LST-4@ZnPc NPs were characterized in terms of morphology, elemental composition, particle size, surface charge, UV–Vis absorption spectra, and fluorescence spectral signals. The samples were dropped onto a carbon-coated copper grid for TEM measurements. The solution of LST-4 or LST-4@ZnPc NPs sample was incubated with acid aqueous solution (pH = 3 or 5) for a period of time, which monitored the changes in UV–Vis absorption and fluorescence. The cargo (ZnPc) concentration in the LST-4@ZnPc NPs was determined spectrophotometrically. All determinations were performed at least three times.

### 3.9. Loading Efficiency (LE) and In Vitro Release Capacity of LST-4@ZnPc NPs

The LE of ZnPc was evaluated by disrupting the structure of LST-4@ZnPc NPs, with the cargo concentration in LST-4@ZnPc NPs determined spectrophotometrically as in [50]. Thus, a sample of the LST-4@ZnPc NPs was diluted with an equal volume of water and five volumes of dimethyl sulfoxide (DMSO), followed by sonicating for 30 min. After that, the absorbances of solutions were measured using a Microplate Reader Spark (Tecan Group AG, Männedorf, Switzerland) with maximal absorbance observed at about 674 nm. Free ZnPc samples dissolved in a 5:1 (*v*/*v*) DMSO:water mixture were prepared to establish a calibration curve.

The LE of ZnPc was determined by the Formula (1):
(1)$$LE\%=\frac{Weight\;of\;ZnPc\;in\;NPs}{Weight\;of\;added\;ZnPc}\times 100\%$$

The release capacity of LST-4@ZnPc NPs was determined using a dialysis method in vitro. Firstly, 1 mL of LST-4@ZnPc NPs stock solution was placed in 14 kDa dialysis chambers and dialyzed against 15 mL of pH 5.0 or pH 7.4 PBS solution with shaking at 37 °C. At various time points, aliquots of 1.0 mL were withdrawn and immediately replaced with the same volume of fresh release media. The ZnPc content in the withdrawn samples was determined by a microplate reader.

### 3.10. In Vitro ^1^O_2_ Detection of LST-4@ZnPc NPs

DPBF is used as a capture agent for ^1^O_2_. LST-4 (or ZnPc) was dissolved in DMSO to prepare a 3 mL test solution with a concentration of 0.2 μM, containing DPBF at a concentration of 20 μM. In the concurrent phase, only a DPBF (20 μM) solution in DMSO was prepared to eliminate potential sequence errors from DPBF. Then, the spectral changes of test solutions would be collected every 15 s for 2 min with a UV-2450 spectrophotometer in the range of 550–300 nm under a 690 nm irradiation. Based on the above, the ^1^O_2_ generation ability of LST-4@ZnPc NPs (or LST-4 NPs) in an aqueous solution containing 20% DMSO was reconfirmed using the same method. 

### 3.11. In Vitro Stability Assay 

The prepared solutions were incubated with water, PBS, RPMI Medium1640, and RPMI Medium1640 containing 50% fetal bovine serum (FBS) at room temperature for 24 h. We centrifuged the samples to check for any precipitation to evaluate the physiological environment stability of the prepared solution. Similarly, the hydrodynamic diameters of samples were tested within 10 days to verify whether the LST-4@ZnPc NPs could exist stably at 4 °C for a long time.

### 3.12. Cell Culture

The HepG2 cell line and L02 cell line were purchased from Shanghai Meixuan Corporation (Shanghai, China). All cell lines were cultured in RPMI Medium1640 containing 10% FBS, 100 μg mL^−1^ streptomycin, and 100 U mL^−1^ penicillin at 37 °C in a humidified incubator containing 5% CO_2_ and 95% air. The medium was replenished every other day and the cells were sub-cultured after reaching confluence.

### 3.13. In Vitro Cytotoxicity Assay 

To investigate the cytotoxicity of LST-1, LST-2, LST-3, LST-4, LST-4 NPs, ZnPc, and LST-4@ZnPc NPs, CCK-8 assays were performed on the HepG2 and L02 cells. Briefly, HepG2 and L02 cells were separately seeded at a density of 8 × 10^3^ cells per well in 96-well plates in 100 μL of complete medium and incubated at 37 °C for 24 h. After rinsing with PBS, the HepG2 and L02 cells were incubated with 100 μL culture media containing serial concentrations of LST-1, LST-2, LST-3, and LST-4 for 24 h, respectively. Additionally, the cell viability was assessed with the CCK-8 assay. Based on the above research, the HepG2 cells were incubated with 100 μL culture media containing serial concentrations of LST-4 NPs, ZnPc, and LST-4@ZnPc NPs for 24 h. The ZnPc and LST-4@ZnPc NP groups included the light or dark groups. Among these, the light groups received laser irradiation at 690 nm (0.2 W/cm^2^) for 5 min. The dark groups were used as the control groups. Ultimately, the HepG2 cells were incubated for an additional 24 h in the dark and the absorbance at 450 nm was measured with a microplate reader to calculate the cell viability. 

### 3.14. Western Blot Analysis 

To examine the effect of LST-4 on the protein expression level in HepG2 cells, the various concentrations (0, 1/2 × IC_50_, IC_50_, 2 × IC_50_) of LST-4 treated cells were harvested and lysed in RIPA lysis buffer for 30 min at 4 °C. The supernatant was obtained by centrifugation at 12,000×*g* for 30 min. The proteins were separated by SDS polyacrylamide gel electrophoresis in 8–12% gels and transferred onto polyvinylidene fluoride at a low temperature. After blocking the membranes for 0.5 h at room temperature in rapid blocking buffer, they were incubated with an appropriate primary antibody (1:1000) at 4 °C for 12 h. The membranes were then washed with tris-buffered saline (TBS) containing 0.1% of tween-20 and probed with secondary antibodies (1:1000) for 1 h at room temperature. After washing, the protein expression was measured.

### 3.15. Detection of Intracellular GSH

To assess the capability of LST-4 and LST-4 NPs in consuming GSH, the GSH and GSSG Assay Kit (Biyuntian Biotechnology Co., Ltd., Shanghai, China) was used to detect intracellular GSH content. Briefly, HepG2 cells were cultured in a 6-well plate (2 × 10^5^) for 24 h, then treated with culture media containing serial concentrations (0, 1/2 × IC_50_, IC_50_, 2 × IC_50_) of LST-4 and LST-4 NPs for 24 h, respectively. The cells were then collected with trypsin and treated according to the kit instructions. Then, various relevant data were measured using a microplate reader, and the GSH content of each group was calculated according to the reagent instructions.

### 3.16. In Vitro Cellular Uptake

HepG2 cells were cultured on confocal dishes for 24 h, after which the cells were treated with LST-4 NPs and LST-4@ZnPc NPs at 0.5 h, 1 h, 2 h, and 4 h, respectively. Then, after washing with PBS three times, the cells were stained with Hoechst 33342 for 15 min. After washing with PBS again three times, the samples were imaged by a Zeiss Axio Observer 7 inverted FL microscope with a 63× oil immersion objective and analyzing by ZEN 3. 1 (v. 3. 1. 0. 00013, Carl Zeiss). The FL emissions of the LST-4@ZnPc NPs were 525 nm (LST-4 NPs) and 683 nm (ZnPc), respectively. The FL emissions of the LST-4 NPs and Hoechst 33342 were 525 nm and 425 nm, respectively.

### 3.17. Detection of Intracellular ROS

To detect intracellular ROS accumulation in living cells, CellROX™ Orange was used as an indicator for ROS in cells to avoid interference from the fluorescence channels of LST-4@ZnPc NPs. The HepG2 cells (about 1 × 10^4^) were incubated with LST-4@ZnPc NPs (20 μg/mL) in a confocal small dish and cultured in an incubator for 24 h, including light and dark groups. After washing once with PBS, CellROX™ Orange (2.5 μL, 20 μM) was added and cultured with the cells for 30 min. The cells were washed again twice to remove extracellular CellROX™ Orange and then irradiated with a 690 nm laser (0.2 W/cm^2^) for 2 min per dish. Eventually, live cell fluorescence imaging was performed with a fluorescent microscope (Axio Observer 7, Zeiss, Oberkochen, Germany). The FL Emission of CellROX™ Orange is 572 nm and Vitamin C (1.0 mM) was used to eliminate cellular ROS as a positive control.

### 3.18. Subcellular Localization

HepG2 cells (about 1 × 10^4^) were cultured on confocal dishes for 24 h, then incubated with LST-4 NPs (20 μg/mL) for 4h. After incubation, the cells were stained with a combination of LysoTracker Red, MitoTracker DeepRed, and Hoechst 33342 for 15min. Then, the cells were washed twice with PBS and again with blank medium to remove additional dyes and nanoparticles. On the other hand, the subcellular localization of LST-4@ZnPc NPs is similar to that of LST-4 NPs. Cells treated with LST-4@ZnPc NPs (20 μg/mL) were stained with a combination of LysoTracker Redand Hoechst 33342 for 15 min. Lastly, fluorescence imaging of the cells was performed using a Zeiss Axio Observer 7 inverted fluorescence microscope with a 63× oil lens. The results were analyzed by the software ZEN 3. 1 (v. 3. 1. 0. 00013, Carl Zeiss). The fluorescence emission wavelengths of Hoechst 33342, LysoTracker Red, MitoTracker Deep Red, LST-4 NPs, and LST-4@ZnPc NPs were 455 nm, 512 nm, 589 nm, 525 nm, 525 nm, and 683 nm, respectively. 

### 3.19. Statistical Analysis

The obtained data were all statistically analyzed using Origin Pro 2024 (v. 10. 1. 0. 178). The experimental data are presented as mean ± standard error (mean ± SEM). Student’s test was used to test the data statistics between two groups, and Dunnett’s test was used after one-way ANOVA when there were three or more groups. *p* < 0.05 is considered to be statistically significant.

## 4. Conclusions

In summary, in this study, we developed a novel functional nanosphere (LST-4@ZnPc NPs) by self-assembling nanocarriers (LST-4 NPs) through a new matrine derivative (LST-4), loading successfully with ZnPc for effectively killing cancer cells via combination therapy including both chemotherapy and synergistic amplification of oxidative damage. LST-4@ZnPc NPs were found to not only release ZnPc in response to the TME to produce abundant ROS upon exposure to light, but also exert the characteristics of matrine to further break through the TME and decrease the intracellular GSH level, thereby triggering high-level synergistic amplification of oxidative damage in tumor cells. Meanwhile, LST-4, as a representative incorporating PROTAC technology into natural products, not only improved biological activity on the basis of preserving its original properties including consuming GSH but also made it possible for natural products that were not suitable for self-assembly to achieve self-assembly. Furthermore, the LST-4 NPs formed through LST-4 self-assembly exhibited a good EPR effect, as their characterization results also indicate that LST-4 NPs can serve as a good delivery platform. Ultimately, LST-4@ZnPc NPs are not only a chemotherapy drug, but also can promote the production of ROS and consume GSH, producing stronger anti-tumor effects. Altogether, the assembled combination based on the active ingredients of traditional Chinese medicine, forming both photosensitizers and functional chemotherapy agents, is a promising method for the treatment of cancer and also provides a new paradigm for using natural products to kill tumors.

## Data Availability

The data are available within this article and its Appendix A.

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
