# Peer review of "Self-Assembled Matrine-PROTAC Encapsulating Zinc(II) Phthalocyanine with GSH-Depletion-Enhanced ROS Generation for Cancer Therapy"

_molecules, 2024, doi:10.3390/molecules29081845_

Round 1

Reviewer 1 Report

Comments and Suggestions for Authors

Please see the attached pdf.

Comments on the Quality of English Language

Most of the 'was' in the manuscript , where the authors describe a phenomena, should be replaced with 'is'. 

Author Response

To Reviewer 1

Comments and Suggestions for Authors

General comments

In the article, the authors describe the synthesis, characterization, and anti-cancerous activities of a series of LST nanoparticles. Furthermore, an LST-4 was functionalized with a zinc phthalocyanine that can possibly enhance the cytotoxicity through combined effect of photo and chemotherapy. The LST molecules have been characterized through 1H-NMR and HRMS, however, the caption of all the NMR figures must mention the solvent used. A sentence in the abstract “Moreover …. Cells” is too long and must be rewritten in 2 to 3 sentences. There are several instances in the manuscript where abbreviations are used (for e.g. LST, EPR, PEG, DPBF, etc.) and their full form should be mentioned when the abbreviations are used for the first time. A few lines on the nature of the interaction or the bonding forces between LST-4 NP

and ZnPc should be discussed in the text.

Response:1. Thank you for your comment. Based on your suggestion, we have added solvents we used of all NMR figures. Supporting Information Figure S1, Figure S2, Figure S3, Figure S4, Figure S5, Figure S6, Figure S7, Figure S8, Figure S9, Figure S11, Figure S13, Figure S15.

  1. Thank you for your suggestion. Based on your suggestion, we have revised the sentence and now the expression is more concise. Page 1, Lines 25-30.
  2. We have provided supplementary explanations for the first occurrence of abbreviations in the entire text and provided a comparison table of abbreviations at the end of the article.
  3. We have provided some explanations of the interaction between LST-4 NPs and ZnPc. Page 6-7, Lines 202-207.

Specific comments

  1. I have some other comments that should be addressed before the article is considered for a publication. ‘Section 2.2’: LO2 should be replaced with L02.

Response:We are very sorry for our lack of rigor. Now it has been corrected.

  1. ‘Section 2.3’: Lines 147-150 should be merged as both convey the same message. Response: We thank the Reviewer good advice. We have removed duplicate semantics to make the writing smoother.
  2. Figure 2 caption: What is the significance of mentioning n=4 or n=3 here?

Response:Thank you for your question. We have repeated parallel experiments to reduce systematic errors through.

  1. ‘Section 2.5’: Please mention the relevant reference for the sentence “Furthermore, numerous … activity”.

Response:Thank you for your advice. We have added relevant literature. Page 6, Line 199.

  1. Line 208, “LST-4 displayed from the side.” meaning? Line 214 (Figures 4d, 4e

and 4f).

Response:Thank you for your patient correction. We are very sorry for our lack of rigor. Due to writing errors, semantic errors were caused. Now it has been corrected. In article, we want to express “ LST-4@ZnPc NPs and LST-4 NPs also exhibited significant redshift in fluorescence and UV-Vis absorption spectra compared to LST-4 monomers (Figure 4b, 4c), which displayed from the side that LST-4 NPs have been successfully synthesized, LST-4@ZnPc NPs loading with ZnPc have been synthesized successfully”. Page7, Lines 221-222.

“Figures 4e, 4f and 4g” has been corrected. Page7, Line 226.

  1. Figure 4 caption: The concentrations of the solutions for UV-vis and FL should be mentioned. ‘Section 2.9’: What was the amount of LST-4@ZnPc NP that was used in the tumor cells for the ROS study? What decides the amount?

Response: Thank you for your question. According to your suggestion, we added concentrations of solutions for UV-vis and FL in the caption of Figure 4. Page8, Lines 249-252.

In the testing of reactive oxygen species, we used the dosage of LST-4@ZnPc NPs is 20ug/mL. We marked this information in page 18, line 650. We decided to use 20ug/mL LST-4@ZnPc NPs for ROS study is because of the IC50 value of LST-4@ZnPc NPs is 23.45 ug/mL after 5 min of irradiation with a 690 nm laser.

  1. Most of the “was” in the manuscript, where the authors describe a phenomena, should be replayed with “is”.

Response: Thank you for your help. We have carefully revised it.

Reviewer 2 Report

Comments and Suggestions for Authors

Please provide in vivo study data.  The current study does not support the conclusion. 

Comments on the Quality of English Language

NA

Author Response

To Reviewer 2

Comments and Suggestions for Authors

  1. Please provide in vivo study data. The current study does not support the conclusion.

Response:Thank you for your comments. We strongly agree that more studies would be useful to understand the details of interaction and enhancement. At this point, we do not have the necessary tool to study vivo study. We hope, in the future, we can explore in this area to make our research more comprehensive.

Reviewer 3 Report

Comments and Suggestions for Authors

The manuscript is well written in detail. The topic is interesting, this can be considered for publication after following modifications:

1. Sections 3.3  3.6  3.7 The synthesis method used in this study is authors own method or they adopted from other research, if from other then provide appropriate reference.

2. There are many abbreviations were used in this study, therefore Author should provide a se list of abbreviations for better understanding.

3. It is suggested to provide comparison of this study with other related research.

Author Response

To Reviewer 3

The manuscript is well written in detail. The topic is interesting, this can be considered for publication after following modifications:

Response:Thank you for your comment.

  1. Sections 3.3 3.6 3.7 The synthesis method used in this study is authors own method or they adopted from other research, if from other then provide appropriate reference.

Response:We thank the reviewer valuable suggestion. According to your suggestion, we provided appropriate references. Page 13, Line 397. Page 16, Line 542.

  1. There are many abbreviations were used in this study, therefore Author should provide a se list of abbreviations for better understanding.

Response:Thank you for your suggestion. Based on your suggestion, we have added an abbreviations comparison table at the end of the article.

  1. It is suggested to provide comparison of this study with other related research.

Response:We thank the reviewer good advice. We have made modifications. We have provided the comparison of this study with other related research. Page 2, Lines 74-81. Oxidation therapy of cancers is an attractive strategy involved in chemotherapy, radiotherapy, and photodynamic therapy. It is a feasible oxidative therapy approach to break the redox state in tumor microenvironment. Many researchers have conducted similar studies, but there are some limitations, such as Ge and co-workers prepared a nanocomplex to enhance chemodynamic therapy by incorporating Fe3O4 and glucose oxidase into a polyprodrug-based vesicule. Lan and his colleagues also constructed carrier-free nanodrug made up of gambogic acid, chlorin e6 and folic acid for cooperative cancer treatment. However, the inorganic or metallic properties of nanomaterials like Fe3O4, as well as the addition of multiple theranostic agents, increase the metabolic load raising concerns about their potential toxicity to normal tissues. Therefore, our study started with the active ingredients of traditional Chinese medicine and then encapsulates ZnPc, which not only increases biological safety to reduce the metabolic load of the body, but also increases the spatial utilization of nanomedicines to amplify oxidative damage to tumors.